# The Impact of a Positive Biofire^®^ FilmArray^®^ Gastrointestinal Panel Result on Clinical Management and Outcomes

**DOI:** 10.3390/diagnostics13061094

**Published:** 2023-03-14

**Authors:** David Carmon, Hanan Rohana, Maya Azrad, Avi Peretz

**Affiliations:** 1Azrieli Faculty of Medicine, Bar-Ilan University, Safed 1311502, Israel; 2The Clinical Microbiology Laboratory, Tzafon Medical Center, Poriya, Tiberias 1528001, Israel

**Keywords:** gastrointestinal infections, FilmArray^®^ Gastrointestinal Panel, pathogen distribution, positive result, clinical management and outcomes

## Abstract

The gold standard diagnostic method for gastrointestinal infections is stool culture, which has limited sensitivity and long turnaround time. Infection diagnosis recently shifted to syndrome-based panel assays. This study employed the FilmArray^®^ Gastrointestinal Panel, which detects 22 pathogens simultaneously, to investigate gastrointestinal infection and pathogen distribution in 91 stool samples of patients hospitalized at the Tzafon Medical Center, Israel, during 2020, and to compare the clinical and demographic data of negative vs. positive samples. Among the 61 positive samples (67%), the most common pathogen was *Campylobacter* (34.4%). Positive test results were associated with a slightly younger patient age (*p* = 0.012), significantly higher post-diagnosis use of antibiotics (63.9% vs. 36.7%; *p* = 0.014), and shorter length of stay and time to discharge (*p* = 0.035, *p* = 0.003, respectively) than negative test results. To conclude, the FilmArray^®^ Gastrointestinal Panel enabled the early identification of causative infectious agents and enhanced clinical management and outcomes.

## 1. Introduction

Gastrointestinal infections (GIs) pose a global threat to public health and are associated with high rates of morbidity and mortality [1]. According to the results of the Global Burden of Disease Study 2016, acute gastroenteritis was responsible for approximately 89.5 million disability-adjusted life years lost and 1.45 million deaths per year [2]. In the United States, about 179 million people suffer from diarrheal illnesses each year [3]. 

GIs have been linked to the development of other health conditions, such as Reiter’s syndrome, Guillain–Barré syndrome and irritable bowel syndrome (IBS) [4,5]. Symptoms may include vomiting, diarrhea, abdominal pain and fever [6].

Given the nonspecific signs presented in GIs, the etiology is often unknown. Primary treatment, based on clinical manifestations, may include the use of antimicrobial agents [7]. However, inappropriate antibiotic use may lead to the emergence of multidrug-resistant organisms, especially among vulnerable populations [7,8]. Furthermore, in viral gastroenteritis or uncomplicated bacterial gastroenteritis, antibiotics can be harmful. For example, in non-typhoid *Salmonella* infections, antibiotics increase the risk of disease relapse and prolonged carriage, and in Shiga-toxin-producing *Escherichia coli (E. coli*) infections, antibiotics elevate the risk of hemolytic uremic syndrome [6]. Therefore, an accurate diagnosis is critical in order to assure etiology-directed treatment regimens and reduce any possible complications.

A wide range of pathogens has been attributed to GI etiology [9,10]. In bacterial gastroenteritis, the gold standard diagnostic method is stool culture [11]. However, this technique is limited due to the fact that some bacteria have non-conventional growth requirements; the time to result is usually long (2–5 days), and inappropriate sample handling is possible and might affect the result, as in cases of collecting the stool sample after antibiotic treatment onset. 

Other conventional diagnostic methods include microscopic examination for parasitic pathogens and enzyme immunoassays (EIAs), which are used to identify viral, bacterial and parasitic infections [11]. 

In recent years, there has been an applicative leap in infectious disease diagnostics, from traditional to syndromic-based diagnosis, using molecular assays that simultaneously detect several pathogens including viruses, fungi, parasites and bacteria within a few hours [12]. One such platform is the Biofire^®^ FilmArray^®^ (BioFire^®^ Diagnostics, Inc., Salt Lake City, UT, USA), which provides different panels for the detection of common pathogens associated with diverse infectious conditions, including respiratory tract infections, meningitis and GIs.

The FilmArray^®^ Gastrointestinal Panel (GIP) has the most comprehensive array of targets (22 in total), checking for bacteria, viruses and parasites known to cause IGE. Different works have shown that the specificity and sensitivity of the GIP are 97.1% and 94.5–100%, respectively [11,13]. The run time is about an hour, and the required amount of sample is small (200 μL). This technology carries out nucleic acid extraction followed by polymerase chain reaction (PCR) in which different DNA targets are amplified and detected [11]. Thus, it screens multiple pathogens within a single test run, has high detection rates and wide detection coverage. However, its main disadvantage is its high cost and the lack of interpretable quantification [11,14].

The current study investigated GI pathogens’ distribution in stool samples of patients who were hospitalized at the Tzafon Medical Center, Poriya, Israel, during 2020. Clinical and demographic data of patients with negative vs. positive GIP were compared and the antibiotic status was thoroughly examined (including changes in antibiotic treatment after the test result and antibiotic treatment at discharge). The main aim of this study was to assess the impact of a positive GIP result on clinical management and outcome. We found that a positive GIP result was significantly associated with a higher post-diagnosis use of antibiotics (63.9% vs. 36.7%), and a shorter length of stay and time to discharge (*p* = 0.035, *p*= 0.003, respectively) than a negative GIP result. Therefore, the early identification of GI pathogens by the FilmArray^®^ Gastrointestinal Panel enhances clinical management and outcomes.

## 2. Materials and Methods

### 2.1. Study Design

This study retrospectively analyzed 91 records of patients, aged 0–100 years, who were admitted to the Tzafon Medical Center, Poriya, Israel, during 2020, with a suspicion of IGE. Stool samples were collected as part of the routine medical care service and tested with the Biofire^®^ FilmArray^®^ GIP following the request of an infectious disease specialist. Clinical and demographic data were collected from patients’ medical records including age, ethnicity, gender, chronic diseases, acute complications, vaccination status (the routinely administered vaccines in Israel), time passed from hospitalization to GIP test, time passed from GIP test result to discharge from hospital, length of stay (LOS), death during hospitalization and whether GIP result has led to discharge. In addition, data concerning antibiotic status were collected including antibiotics administration before and after test result, changes in antibiotic treatment following the test result and treatment on discharge. The study was approved by the institutional review board (IRB) of the Tzafon Medical Center, approval no. POR-0105-20. The IRB waived the need for informed consent.

### 2.2. Sample Collection and Biofire^®^ FilmArray^®^ GIP Test

Stool samples were collected and sent to the clinical microbiology laboratory at the medical center. Each sample was tested by the Biofire^®^ FilmArray^®^ GIP according to the manufacturer’s instructions. Briefly, stool samples were transported into Cary Blair transport media (FecalSwab, Copan Diagnostics Inc., Murrieta, CA, USA) and the tube was inverted several times. Afterwards, a hydration solution was injected into the FilmArray^®^ GI pouch, 200 μL of sample were mixed with the provided buffer and the mixture was injected into the test pouch (provided with all necessary reagents in a freeze-dried state). The pouch was inserted into the instrument. At the end of each run (~1 h per each sample), results were presented on the computer’s software; when pathogen was detected, the software introduced the pathogen name on the computer screen. When no pathogen was detected, a “Negative” result was presented. A positive result was reported to the clinical staff by phone. 

The Biofire^®^ FilmArray^®^ system detects seven bacteria: *Campylobacter jejuni* (*C. jejuni*), *Campylobacter Coli* (*C. coli*), *Campylobacter upsaliensis* (*C. upsaliensis*), *Clostridiodes difficile* (*C. difficile*), *Plesiomonas shigelloides* (*P*. *shigelloides*), *Salmonella* spp., *Yersenia enterocolitica* (*Y. enterocolitica*), *Vibrio parahaemolyticus* (*V. parahaemolyticus*), *Vibrio vulnificus* (*V. vulnificus*), and *Vibrio cholera* (*V. cholera*); six diarrheagenic *Shigella* spp./*E. coli*: enteroaggregative *E. coli* (*EAEC*), enteropathogenic *E. coli* (*EPEC*), enterotoxigenic *E.coli* (*ETEC*), Shiga-like toxin producing *E. coli* (*STEC*), *E. coli O157*, and enteroinvasive *E. coli* (*EIEC*)/*Shigella* spp.; four parasites: *Cryptosporidium*, *Cyclospora cayetanensis* (*C. cayetanensis*), *Entamoeba histolytica* (*E. histolytica*), and *Giardia lamblia* (*G. lamblia*); and five viruses: Adenovirus F 40/41, Astrovirus, Norovirus GI/GII, Rotavirus A and Sapovirus [13].

### 2.3. Stool Culture

Stool samples were cultured as part of the routine GI diagnosis in Israel, which included testing for *Salmonella*, *Shigella* and *Campylobacter*. To this end, all study specimens were inoculated on Xylose Lysin Deoxycholate (XLD) agar (Hylabs Ltd., Rehovot, Israel) and *Salmonella Shigella* (SS) agar (Hylabs Ltd.) and incubated at 37 °C for the detection of *Salmonella* and *Shigella*. Samples were also inoculated on *Campylobacter* CVA agar (Hylabs Ltd.) and incubated at 42 °C under microaerophilic conditions for the detection of *Campylobacter*. After 48 h incubation, suspicious colonies were further identified using the Matrix-Assisted Laser Desorption Ionization–Time of Flight (MALDI–TOF) technique by the Bruker Biotyper system (Bruker Daltonics, Bremen, Germany) [15].

### 2.4. Statistical Analysis

The study patients were grouped according to their GIP result into positive (GIP^+^) and negative (GIP^−^) groups. All measured variables and derived parameters were tabulated by descriptive statistics. Chi-squared test was applied for assessing the difference in categorial variables between the two groups. The two-sample *t*-test for independent groups was applied for testing the statistical significance of the difference in quantitative measurements between the study groups. 

All tests were two-tailed, and a *p*-value of 5% or less was considered statistically significant.

The data were analyzed using the R ^®^ software 9.4 (R score team, 2022).

## 3. Results

This study included records of 91 patients; 61 (67%) had a positive GIP result and 30 (33%) had a negative result. 

### 3.1. Distribution of Causative Agents

A total of 88 pathogens were detected in the 61 positive samples. Among the 61 positive results, 43 (70.5%) were attributed to bacterial etiology with the most common bacteria being *Campylobacter* (21/43, 48.8%) and *Salmonella* (15/43, 34.9%) (Figure 1); 17 (27.9%) patients had a viral etiology, with the most common viruses being Rotavirus (9/17, 52.9%) and Sapovirus (5/17, 29.4%); and only one parasite (1/61, 1.6%) was detected—*G. lamblia*.

### 3.2. Comparison of Culture and GIP Test Results

We found a significant association between the GIP result and the stool culture result (*p* = 0.0176). Within the GIP^+^ group, *Campylobacter* was detected in 12 (19.7%) specimens and *Salmonella* in 9 (14.7%). No pathogens were detected in stool cultures within the GIP^−^ group. Blood cultures were performed for 68 patients. Within the GIP^+^ group, one sample was positive for *Salmonella*.

### 3.3. Coinfections

Of the 61 GIP^+^ group patients, 23 (37.7%) had multiple pathogens detected, with either 2, 3 or 4 pathogens detected in 20 (32.8%), 1 (4.3%) and 1 (4.3%) of the positive specimens, respectively (Figure 2). 

The most common pathogens detected as coinfections were *EAEC* and *EPEC*, found in 4 (17.4%) specimens, followed by *Campylobacter* and *EAEC* in 3 samples (13%). 

### 3.4. Distribution of Causative Agents, Per Age Groups 

The analysis of pathogens by age group showed that 21 (34.4%) patients in the GIP^+^ group were under the age of 1 year, 23 (37.7%) were in the range of 2–5 years, 10 (16.4%) were between 6 to 17 years, 4 (6.5%) were older than 18 years and younger than 65 (18–64), and 2 (3.3%) were older than 65 (65+) (Figure 3).

Within the 0–1 group, the most common pathogen was *Campylobacter* (11/21, 52.4%), followed by Rotavirus A (4/21, 19%). Similarly, *Campylobacter* was the most prevalent agent within the 2–5 group (8/23, 34.8%), followed by *Salmonella* (6/23, 26.1%) and *EAEC* (6/23, 26.1%). Within the 6–17 group, the most common pathogen was *Salmonella* (4/10, 40%).

### 3.5. General Demographic and Clinical Data of Patients with a Positive Result vs. a Negative Result of the Gastrointestinal Panel

No significant differences were found between the positive and negative GIP groups in terms of ethnicity, gender, vaccination status and chronic disease (Table 1). Thirty-one (50.8%) patients in the GIP^+^ group were males, 60.7% were Arabs, 96.7% were vaccinated, and 8.2% had a chronic disease. In the GIP^−^ group, most (60%) patients were female, 56.7% were Jewish, and 10% had a chronic disease. 

In contrast, we found a significant association between the GIP result and the patient’s age, with the GIP^+^ patients being a bit younger (*p* = 0.012). The mean age of the GIP^+^ was 7.3 ± 15.8 years while the mean age of GIP^−^ was 18.9 ± 27.4 years.

### 3.6. Antibiotic Status, Length of Stay and Outcomes

A significantly higher use of antibiotics after the GIP test result (*p* = 0.014) was found among patients from the GIP^+^ group (63.9%) compared to GIP^−^ (36.7%) (Table 2). No significant difference was found between the two groups in terms of antibiotic use prior to testing, nor in terms of change in antibiotics/discontinuation of antibiotics treatment after test. 

The time to sample was not significantly different between the GIP^+^ and GIP^−^ groups; however, both the time to discharge and duration of hospitalization were significantly higher in the GIP^−^ group compared to GIP^+^ (*p* = 0.035, *p*= 0.003, respectively).

Furthermore, the treatment received on discharge was more common in the GIP^+^ group (50.8%) compared to the GIP^−^ group (26.7%) (*p* = 0.029). 

## 4. Discussion

The current main GI etiology detected by the GIP in the current study was bacterial (70.5%), with the most common being *Campylobacter* (48.8%) and *Salmonella* (34.9%). Both pathogens play an essential role in GIs, as reported in other studies [10,16]. The incidence of *Campylobacter* infections has been increasing worldwide along the years and it is considered a leading cause of bacterial GIs [17]. According to a recent report of the Foodborne Diseases Active Surveillance Network in the USA (FoodNet), during 2019, the incidence of *Campylobacter* spp. per 100,000 population was the highest (19.5%), followed by *Salmonella* (17.1%) [18]. In a study from Washington, *Campylobacter* spp. and *Salmonella* spp. were very common (20.9% and 12.4%, respectively); however, the most common pathogen detected was *C. difficile* (55%), compared to 6.6% in our study [19]. This may result from different stewardships of the GIP usage. In our medical center, the use of the GIP is authorized following a consult with an infectious disease specialist. Additionally, *C. difficile* presence is tested via another molecular platform, the GeneXpert^®^ system (Cepheid, Sunnyvale, CA, USA). Thus, stool samples of patients suspected of *C. difficile* infections will probably not be tested by the GIP, and this may explain the low rate (6.6%) of *C. difficile* in our study compared to other studies (51.1–55%) [19,20]. Another factor that affects *C. difficile* prevalence is the patient’s age; while *C. difficile* is more common in the elderly, *Campylobacter* and *Salmonella* are common in all ages, but particularly in children [21,22]. The average age in our study was 11.2 years, as compared to 57 years [20] and 47.4 years [19]. Therefore, the higher rates of *C. difficile* in these previous studies are reasonable.

The routine diagnosis of stool culture in clinical microbiology laboratories in Israel constitutes testing for three bacteria: *Campylobacter*, *Salmonella* and *Shigella*. In the current study, *Campylobacter* was detected by stool culture in 19.7% of specimens, compared to 34.4% in the GIP; *Salmonella* was detected by both culture and the GIP in 14.7% of the positive samples. No *Shigella* was detected by stool culture, and only one case (1.6%) was detected by GIP. These results reinforce previous reports, according to which the GIP has a higher positivity rate compared to culture [19,23,24,25,26]. This can be attributed to the main limits of culture, including the difficulties in culturing bacteria when the sample was collected after antibiotic administration or when the causative bacterium is fastidious and has special growth requirements. Ours and previous results highlight the importance of incorporating molecular methods as part of the routine testing of stool specimens in microbiology laboratories.

Of the 61 positive GIPs, 37.7% had multiple pathogens detected. Previous studies have reported on various rates of coinfections being in the range of 10.2–51.7%. The notable coinfection rate sheds light on the challenge of determining the predominant agent in a considerable number of cases. Such cases require further investigations, as some pathogens may be regarded as irrelevant; for example, the presence of *C. difficile* in infants and young children is thought to be asymptomatic colonization [27]. Another example is the long shedding of the pathogen, such as in cases of *Salmonella* and Norovirus [13], and thus a former ancient infection should be taken into consideration. Since the GIP lacks quantification, it is important to promote multiplex molecular assays that quantify the pathogens involved in GIs, especially when more than one pathogen is detected, as this can distinguish infection from carriage. Additionally, the evidence of high coinfection percentages indicates that stool culture should be used as a complementary tool in some cases. 

The analysis of pathogens by age group showed that *Campylobacter* was the most common pathogen among children in their first year of life (52.4%) and in the 2–5 years group (34.8%). Similar findings were reported in a previous study according to which *Campylobacter* caused the highest associated burden of diarrhea in 0–1-year-old children from Loreto, Peru and Venda, and South Africa [28]. Generally, certain age groups including children below the age of 4 and people older than 75 years are more vulnerable to *Campylobacter* infections [20]. Risk factors to gastroenteritis in young children include high exposure to outsourced food services, in which food handling, storage and cooking might sometimes be improper [22]. 

In contrast to our findings, EPEC was the main pathogen (23%) in a study from France with 172 children (median age, 1.10 years) [26]. However, atypical EPEC strains found in both diarrheal and asymptomatic individuals cast doubt on EPEC’s role in human infections [29]. It should be noted that while the prevalence of *Campylobacter* infections is well studied in developed countries [30,31], less data exist regarding these infections’ burden in developing countries, apparently due to the inconsistency and low sensitivity of culture; thus, it is possible that the contribution of *Campylobacter* to GI is underestimated [32].

Comparison of the positive and negative GIP groups found no significant differences between the patients except for the patient’s age, with the GIP^+^ patients being younger. This result is not surprising as children, especially under the age of 5 years, are one of the most affected groups [33]. 

Concerning the antibiotic status, a significantly higher use of antibiotics after the GIP test result (*p*-value = 0.014) was found in the GIP^+^ group compared to the GIP^−^ group. This result is reasonable, given the fact that 71.7% of GIP^+^ results were attributed to bacterial etiology. A similar observation was reported in a previous study from Columbia, in which 80.1% of the patients with a positive GIP result started an antibiotic treatment following the test result, compared to 62.5% of patients with a negative GIP result (*p* < 0.00001) [20].

Although no significant difference between the groups was seen in terms of antibiotic change/discontinuation, the use of the GIP seems to be worthwhile, as treatment was changed/discontinued for a considerable number of patients in both groups (50% and 36% in GIP^−^ and GIP^+^, respectively). Previous studies linked a significant increase in targeted antibiotics and decreased use of unnecessary antibiotics to the use of the GIP [34,35,36].

Another parameter, which did not significantly vary between GIP groups but is worth considering, is time to discharge; GIP^+^ patients were discharged in a shorter time compared to the GIP^−^ group (1.5 vs. 3 days, respectively). This outcome may be linked to previous studies’ findings, according to which the use of the GIP, instead of the routine culture method, allowed the removal of patients from unnecessary isolation [37,38,39]. In this regard, it should be emphasized that a negative GIP result is also valuable and influences clinical management.

The duration of hospitalization was significantly lower (*p* = 0.003) in the GIP^+^ group compared to GIP^−^. Moreover, change in diagnosis following GIP test result and treatment received on discharge were significantly higher within the GIP^+^ group (*p* = 0.036, 0.029, respectively). Torres-Miranda et al. found a shorter LOS among patients whose samples were detected by the GIP, compared to patients in the period before using the GIP. Additionally, an earlier initiation of optimal antibiotics was seen in the GIP group [19]. Thus, all of these results, once again, highlight the gains of using the GIP in comparison to conventional stool culturing.

The study’s main limitation is the small sample size; further studies with a larger number of samples should be performed in order to ultimately determine the benefits of GIP use. 

## 5. Conclusions

The GIP offers syndromic testing in a much shorter time compared to traditional methods. The GIP reduces antibiotic misuse and length of stay, which both indirectly decrease healthcare-associated costs and enhance the patient’s clinical management. Additionally, a positive GIP result facilitates and expedites the decision-making process by the physicians, including the determination of whether to stop or change antibiotic treatment, choosing the targeted antibiotic, and introducing/removing a patient from isolation, which also has important implications for infection control. Overall, the introduction of the FilmArray^®^ GIP to the clinical laboratory presents a remarkable advance for the diagnosis of IGE.

## Figures and Tables

**Figure 1 diagnostics-13-01094-f001:**
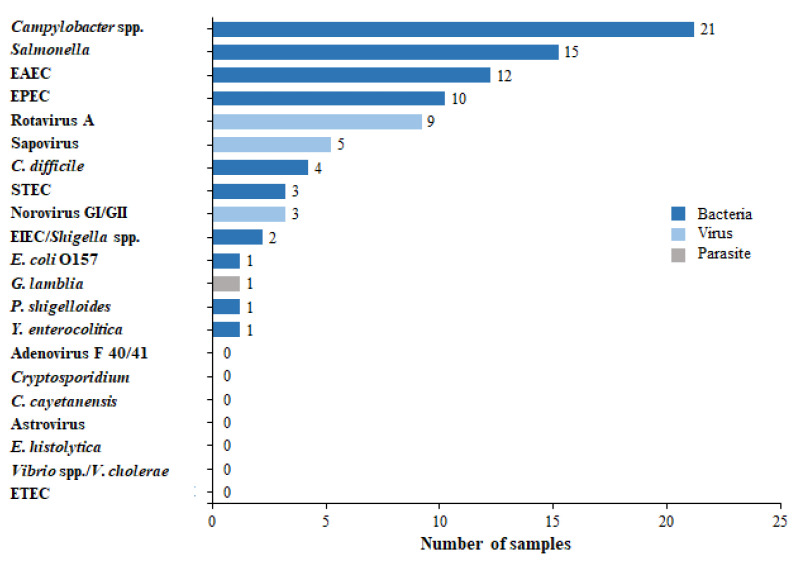
Distribution of causative pathogens in the study group by GIP. Sixty-one stool samples were found to be positive by the GIP for one or more pathogens. The distribution of the different detected pathogens is presented.

**Figure 2 diagnostics-13-01094-f002:**
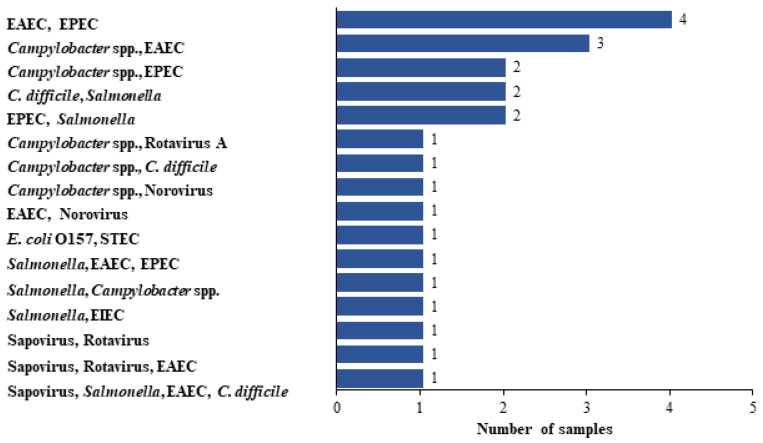
Coinfections detected in GIP-positive specimens. Among the 61 positive stool samples, 24 samples were positive for more than one pathogen (i.e., coinfection), with various combinations of pathogens, as presented in the graph.

**Figure 3 diagnostics-13-01094-f003:**
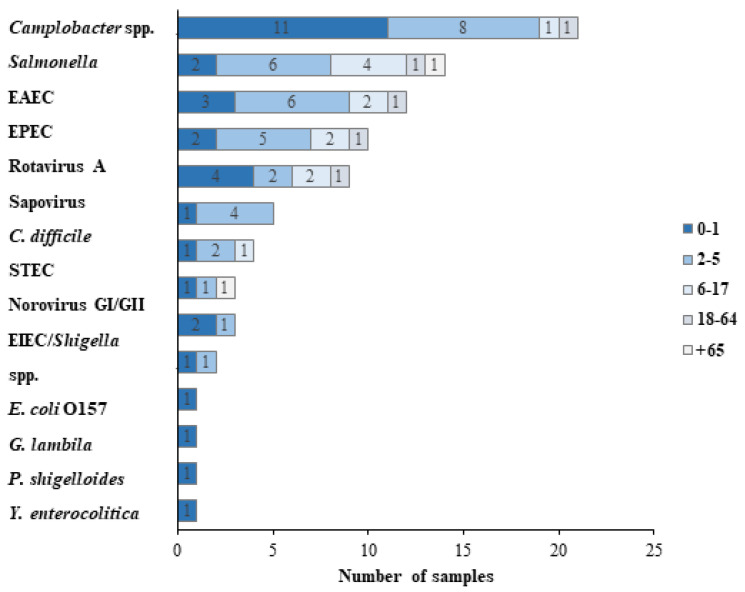
Distribution of causative pathogens in the study group according to age. The various causative agents of GIs distribute differently in distinct age groups, as presented in the graph.

**Table 1 diagnostics-13-01094-t001:** General demographic and clinical data of study participants.

Characteristics	GIP^−^ (N = 30) n, %	GIP^+^ (N = 61) n, %	Total (N = 91) n, %	*p*-Value
Gender, n (%)				0.331
Female Male	18 (60) 12 (40)	30 (49.2) 31 (50.8)	48(52.7) 43 (47.3)	
Age (years)				<0.012
Mean (SD) Range	18.9 (27.4) 0–80.4	7.3 (15.8) 0–76.6	11.2 (20.6) 0–80.4	
Ethnicity, n (%)				
Arab Jews	13 (43.3) 17 (56.7)	37 (60.7) 24 (39.3)	50 (55) 41(46)	0.118
Vaccinated ^1^, n (%)				1.000
No Yes	1 (3.3) 29 (96.7)	2 (3.3) 58 (96.7)	3 (3.3) 87 (96.7)	
Chronic Disease ^2^, n (%)				0.775
No Yes	27 (90) 3 (10)	56 (91.8) 5 (8.2)	83 (91.2) 8 (8.8)	

^1^ Vaccinated refers to vaccination with the routinely administered vaccines in Israel. ^2^ Chronic disease refers to chronic diseases the patient has, such as cancer.

**Table 2 diagnostics-13-01094-t002:** Antibiotic status, length of stay and outcomes.

Parameter	GIP^−^ (N = 30) n, (%)	GIP^+^ (N = 61) n, (%)	Total (N = 91) n, (%)	*p*-Value
Antibiotic treatment prior test, n (%)				0.599
No	11 (36.7)	19 (31.1)	30 (33.0)	
Yes	19 (63.3)	42 (68.9)	61 (67.0)	
Antibiotic treatment after test, n (%)				
No	19 (63.3)	22 (36)	41 (45)	0.014
Yes	11 (36.7)	39 (64)	50 (55)	
Antibiotic change/discontinuation, n (%)				0.203
No	15 (50)	39 (64)	54 (59.3)	
Yes	15 (50)	22 (36)	37 (40.7)	
Time to sample (Days)				0.055
Mean (SD)	1.3 (1.4)	0.7 (1.2)	0.9 (1.3)	
Range	0.0–5.0	0.0–8.0	0.0–8.0	
Time to discharge (Days)				0.035
Mean (SD)	3.0 (4.5)	1.5 (1.8)	2.0 (3.0)	
Range	0–18	0–13	0–18	
Length of stay (Days)				0.003
Mean (SD)	4.1 (4.8)	2.0 (1.5)	2.0 (3.0)	
Range	0–21.0	0–8.0	0–18	
Discharge due to GIP result, n (%)				0.604
No	29 (96.7)	60 (98.4)	89 (97.8)	
Yes	1 (3.3)	1 (1.6)	2 (2.2)	
Acute complications ^1^, n (%)				0.152
No	29 (96.7)	61 (100)	90 (99)	
Yes	1 (3.3)	0 (0)	1 (1)	
Treatment received on discharge, n (%)				0.029
No	22 (73.3)	30 (49.2)	52 (57)	
Yes	8 (26.7)	31 (50.8)	39 (43)	
Change in diagnosis following test, n (%)				0.036
No	29(96.7)	49(80.3)	78 (85.7)	
Yes	1 (3.3)	12 (19.7)	13 (14.3)	

^1^ Acute complications include dehydration, malabsorption, transient lactose intolerance, systemic infection, and sepsis.

## Data Availability

The data used and/or analyzed during the current study are available from the corresponding author on reasonable request.

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
