# Peer review of "The Impact of a Positive Biofire® FilmArray® Gastrointestinal Panel Result on Clinical Management and Outcomes"

_diagnostics, 2023, doi:10.3390/diagnostics13061094_

Round 1

Reviewer 1 Report

I guess the presented paper contain some significantly important data. However, the conclusion needs to be improved to elaborate more the interest to the scientific community and the benefits from using Biofire® FilmArray® over other methods. 

Reviewer 2 Report

The current article (diagnostics-2236408) highlighted the use of FilmArray® Gastrointestinal Panel analysis for more accurate and quick diagnosis of pathogens associated gastrointestinal diseases which could enhanced clinical management and outcome of GI diseases. Though the study is interesting for scientific community and general public but the following comments should be properly addressed.

Comments:

  • Page 2, line 81: age 0-100 may be changed to age 0-100 Years.
  • Page 2, line 82:  Poriya, Israel may be added to Tzafon Medical Center.
  • What was the inclusion and exclusion criteria for selection of patients?
  • Figure 3: The X-axis may be labelled as Age (yr).
  • The spelling of G. lambila may corrected to G. lamblia throughout the manuscript.
  • Table 1: The author refers to which Chronic Disease and vaccination? For better understanding of the readers the Chronic Disease and vaccination should be defined in the table foot note.
  • Table 2: Acute complications may be defined.
  • The limitation of the study should be properly described in the discussion part.
